Detecting temporal asymmetry after epilepsy surgery: a 3D MRI-based comparative outcome study of clinicians and lay observers

Denadai Rafael denadai.rafael@hotmail.com 1
Alvim Marina Koutsodontis Machado 1 2
Kang Yeonah 3
Tu Junior Chun-Yu 4
de Campos Brunno M. 1 5
Ghizoni Enrico 1 2
Tedeschi Helder 1 2
Yasuda Clarissa 1 2 5
Cendes Fernando 1 2 5
1 Brazilian Institute of Neuroscience and Neurotechnology (BRAINN), Universidade Estadual de Campinas , Campinas , São Paulo , Brazil
2 Department of Neurology, School of Medical Sciences, Universidade Estadual de Campinas , Campinas , São Paulo , Brazil
3 Department of Radiology, Seo II Medical Group Clinic , Busan , Republic of South Korea
4 Department of Plastic and Reconstructive Surgery and Craniofacial Research Center, Chang Gung Memorial Hospital , Taoyuan , Taiwan
5 Neuroimaging Laboratory, Universidade Estadual de Campinas , Campinas , São Paulo , Brazil
Gkantidis Nikolaos
Electronic publication date: 2025 Oct 30
Publication date: 2025
Volume: 13
Electronic Location ID: e20201
Received 2025 Mar 6; Accepted 2025 Sep 16
Copyright: ©2025 Denadai et al.
Copyright year: 2025
Copyright holder: Denadai et al.
License: This is an open access article distributed under the terms of the Creative Commons Attribution License, which permits unrestricted use, distribution, reproduction and adaptation in any medium and for any purpose provided that it is properly attributed. For attribution, the original author(s), title, publication source (PeerJ) and either DOI or URL of the article must be cited.
License URL: https://creativecommons.org/licenses/by/4.0/

Keywords: Epilepsy, Asymmetry, Outcome study, Clinicians, Laypeople, Three-dimensional, Diagnostic imaging, Temporal asymmetry

Funding: Coordenação de Aperfeiçoamento de Pessoal de Nível Superior—Brasil (CAPES)—Finance Code 001 Coordenação de Aperfeiçoamento de Pessoal de Nível Superior—Brasil (CAPES) This study was financed by the Coordenação de Aperfeiçoamento de Pessoal de Nível Superior—Brasil (CAPES)—Finance Code 001. Rafael Denadai’s PhD Scholarship was funded by the Coordenação de Aperfeiçoamento de Pessoal de Nível Superior—Brasil (CAPES). There was no additional external funding received for this study. The funders had no role in study design, data collection and analysis, decision to publish, or preparation of the manuscript.

==============================
Background

Resective surgery through pterional approach is an effective treatment for drug-resistant temporal lobe epilepsy, but it may lead to temporal asymmetry in the craniofacial region. Nonetheless, recent systematic reviews showed that there is no gold standard method for the discrimination of a clinically visible abnormal temporal asymmetry from a normal fluctuating asymmetry. This study compares the ability of trained and untrained observers to detect temporal asymmetry and establishes a threshold for clinically detecting abnormal asymmetry.

Methods

Standardized magnetic resonance imaging (MRI)-derived three-dimensional (3D) frontal views of adult patients who underwent temporal lobe epilepsy surgery were used to create a continuum spectrum of preoperative (n = 96) and 12-month postoperative (n = 96) craniofacial images. A panel of 32 untrained lay observers (family members and laypeople) and 32 trained clinicians (surgeons and clinical specialists) independently appraised randomly displayed 3D craniofacial soft-tissue images to assess the presence or absence of temporal asymmetry. Objective linear quantifications of temporal thickness differences were obtained from each preoperative and postoperative MRI scan to identify a potential threshold for subjective asymmetry detection. Temporal asymmetry was further categorized into severity levels I to IV based on incremental asymmetry values.

Results

The temporal thickness difference was significantly (P < 0.001) greater in postoperative images (18.3% ± 9.6%) compared to preoperative images (4.6% ± 1.9%). As temporal asymmetry increased from Level I to IV, a significantly higher proportion of 3D craniofacial images were perceived as asymmetrical by all observer categories (P < 0.001). Both trained clinicians and untrained observers—including surgeons, clinical specialists, family members, and laypeople—demonstrated increased (P < 0.001) detection rates with increasing asymmetry severity. A temporal thickness difference exceeding 10% was clinically detected with over 90% accuracy across all observer categories (P < 0.001), establishing a 10% threshold for the clinical perception of temporal asymmetry.

Introduction

Perfect symmetry does not exist in humans; instead, craniofacial asymmetry is viewed as a continuum, from normal craniofacial fluctuating asymmetry in healthy individuals (Crins-de Koning et al., 2025) to pronounced congenital, developmental, and acquired deformities (Cheong & Lo, 2011; Wenger, Gallagher & Bhoj, 2019). Among craniofacial subunits, the frontally visible temporal region plays a key role in facial symmetry, as its contour—positioned above the zygomatic arch and behind the lateral orbital rim—defines the upper facial width (Vaca et al., 2017; Shay et al., 2022). Unilateral changes in this area can result in noticeable asymmetry (Santiago et al., 2018; Thiensri, Limpoka & Burusapat, 2020).

In this setting, resective epilepsy surgery via the pterional approach is a notable contributor to asymmetrical temporal deformities, due to epilepsy’s global prevalence (Yang et al., 2024), the need for surgery in drug-resistant epilepsy (Pang et al., 2025), and potential postoperative complications (Yasuda et al., 2010; Gonzalez-Martinez, 2025). These deformities often lead to appearance-related preoccupation, disrupting quality of life, self-perception, daily functioning, and social interactions (Klassen et al., 2021), underscoring the need for greater clinical and research attention.

The amount of craniofacial asymmetry reported in both healthy individuals and affected patients varies depending on the assessment method used (Nguyen et al., 2024; Harripershad et al., 2025; Lin et al., 2025; Nishimura et al., 2025). Panel assessment, a common indirect method for evaluating craniofacial asymmetry, relies on subjective judgments from lay observers and clinicians (Zhu, Jayaraman & Khambay, 2016; Tan et al., 2021), with clinicians generally considered more sensitive, though studies report mixed findings on their relative accuracy (Lee, Dumrongwongsiri & Lo, 2019; Tan et al., 2021; Zhang et al., 2023). This approach is widely used to detect visible asymmetries and establish recognition thresholds across various craniofacial subunits, which differ by region (Wang et al., 2017; Lee, Dumrongwongsiri & Lo, 2019). For instance, occlusal canting under 3° often goes unnoticed, while deviations over 4° are detected by 90% of observers (Padwa, Kaiser & Kaban, 1997). However, formal data specific to the temporal region is lacking.

Imaging modality can also influence asymmetry-focused analyses (Lo & Lin, 2023). Subjective and objective craniofacial assessments are limited when based on two-dimensional images, which are prone to distortion, magnification errors, and lack of depth representation (Hsu et al., 2020). The rise of three-dimensional (3D) imaging has improved accuracy, offering undistorted, lifelike representations of the complex craniofacial structure (Wu et al., 2019). Among various 3D reconstruction methods, magnetic resonance imaging (MRI) offers non-invasive, radiation-free, high-contrast, high-resolution soft tissue imaging without visual distractions such as skin color, hair, or facial features that could bias perception (Vander Pluym et al., 2007; Villavisanis et al., 2024). MRI is essential for diagnosing and managing epilepsy, offering high-quality data storage (Biagioli et al., 2025). Neuroimaging repositories like Brazilian Institute of Neuroscience and Neurotechnology (BRAINN) house MRI datasets that have supported epilepsy research and can also contribute to craniofacial studies using images originally acquired for other neuroscience purposes (De Souza et al., 2020; Giacomini et al., 2020).

Recent systematic reviews confirm that no standardized method for analyzing the temporal region has been established (Wang et al., 2017; Gonçalves et al., 2021; Shay et al., 2022; Nasim et al., 2024). Although several grading systems have been proposed, their validity and consistency remain insufficiently examined (Vaca et al., 2017; Wang et al., 2017; Choi et al., 2018; Kim et al., 2018; Huang et al., 2018; Laloze et al., 2019; Gonçalves et al., 2021; Shay et al., 2022; Nasim et al., 2024). Moreover, reliable assessments of unmanipulated 3D images for detecting temporal asymmetry after epilepsy surgery remain lacking. Better understanding how laypeople and clinicians perceive such asymmetry could improve deformity detection, treatment planning, and research design.

Using MRI-derived 3D craniofacial soft-tissue image datasets from adult patients treated with resective surgery for temporal lobe epilepsy, this study aimed to assess the subjective recognition of temporal asymmetry by comparing evaluations from trained clinicians and lay observers. Additionally, the study aimed to determine the threshold value at which a temporal asymmetry is clinically recognized as abnormal.

Materials & Methods

This methodological cross-sectional investigation (Fig. 1) analyzed a standardized craniofacial image dataset obtained from the BRAINN database following approval by the Institutional Review Board (CAAE: 93412318.0.0000.5404; University of Campinas (UNICAMP), Campinas, São Paulo, Brazil) and in compliance with the Declaration of Helsinki. All patients provided written informed consent prior to undergoing MRI scans. The study included MRI-derived images from Brazilian adult patients (n = 96; 53 females; aged 42.6 ± 11.2 years; 58 left-side temporal lobe epilepsy) who consecutively underwent unilateral resective surgery via a modified pterional approach for drug-resistant epilepsy between 2016 and 2022, with a minimum postoperative period of 12 months. The 12-month postoperative evaluation was selected to ensure complete scar maturation, resolution of soft-tissue swelling, and stabilization of surgery-induced change (Bond et al., 2008). All included patients had a confirmed diagnosis of temporal lobe epilepsy, established by one of two experienced epileptologists at the Epilepsy Clinic, UNICAMP, Brazil. Exclusion criteria (n = 11) included the presence of other pathological conditions affecting the head or face (e.g., dento-skeletofacial deformities, prior reconstructive procedures, craniofacial trauma, or surgery) and incomplete datasets.

Figure 1 Flowchart of the cross-sectional study design, including pre- and post-epilepsy surgery magnetic resonance imaging -derived three-dimensional (3D) images.

These images were used for subjective panel assessments by trained clinicians (clinical and surgical observers) and untrained observers (laypeople and family members), objective linear quantification of temporal thickness differences, and the threshold-based detection of asymmetry (represented by the dotted arrow).

The imaging databank was compiled for diagnostic and follow-up purposes, with written patient consent (Alvim et al., 2016; De Souza et al., 2020; Giacomini et al., 2020). Under standardized conditions previously described by our team, all pre- and postoperative MRI images were acquired (3 Tesla Achieva-Intera Philips®: T1-weighted images with isotropic voxels of one mm, slice thickness one mm, no gap, flip angle = 8°, TR = 7.0 ms, TE = 3.2 ms, matrix = 240 × 240, and field of view = 240 × 240) (Fonseca et al., 2012; Coan et al., 2014; Campos et al., 2015; Alvim et al., 2016; De Campos et al., 2016) and preprocessed using a multi-software workflow, with intensity inhomogeneity corrected by the N4 bias field correction algorithm in 3D Slicer and intensity normalization achieved through global scaling and alignment to tissue probability maps during spatial normalization in SPM8 (Ashburner & Friston, 2005; Ashburner, 2007; Avants, Nick & Gang, 2009; Tustison et al., 2010; Fedorov et al., 2012; Chakravarty et al., 2011; Madan, 2015; Alvim et al., 2016; Yushkevich et al., 2016; Villavisanis et al., 2024). Postoperative MRI images were rigidly co-registered to the preoperative images using the General Registration (BRAINS) module in 3D Slicer (Johnson, Harris & Williams, 2007; Kikinis, Pieper & Vosburgh, 2014), ensuring anatomical correspondence for asymmetry assessment without altering local tissue morphology (Friston et al., 1995; Ashburner & Friston, 1997; Ashburner & Friston, 2005).

A combination of advanced image analysis and engineering software was utilized for image processing and measurement: Avizo (FEI, Mérignac, France), Geomagic (3D Systems, Rock Hill, S.C.), and SimPlant O&O (Materialize, Leuven, Belgium). All anatomical landmarks, head orientation reference frames, measurement reference planes, severity stratification levels (I to IV), potential cutoff value for detecting asymmetrical deformities of the temporal region, subjective panel assessments (including 3D image frontal views, task time, and binominal scale system), and objective linear measurement (temporal soft tissue thickness difference in percentage) were standardized according to prior studies (Padwa, Kaiser & Kaban, 1997; Vander Pluym et al., 2007; Stephan & Devine, 2009; Kim et al., 2018; Wu et al., 2019; Hsu et al., 2020; Wan, Tsai & Lo, 2021; Morandi et al., 2022; Kurniawan et al., 2024).

Objective analysis

Using predefined anatomical landmarks and structures of interest—including the lateral orbital rim, temporal bone surface, and external skin surface—identified and interactively verified in the axial, coronal, and sagittal imaging planes, objective measurements of temporal thickness differences (Fig. 2A) were collected from each preoperative and postoperative MRI scan (Vander Pluym et al., 2007; Stephan & Devine, 2009; Bu et al., 2010; Welling et al., 2015; Wysong et al., 2013; Wysong et al., 2014; Kim et al., 2018; Morandi et al., 2022).

Figure 2 Magnetic resonance imaging (MRI)-derived images.

(A) Axial magnetic resonance imaging (MRI) view showing the internal (yellow) and external (green) reference lines used for objective linear distance measurements on the operated (blue) and non-operated (red) sides. The internal line (yellow) was defined from the surface of the non-operated lateral orbital rim to the tangential point on the surface of the non-operated temporal bone. The external line (green) was drawn parallel to the internal reference line along the skin surface. Temporal thickness was measured perpendicularly (blue and red lines) as the distance between the internal reference line (yellow) and the skin surface line (green). (B) Three-dimensional MRI-based craniofacial model in a frontal view, illustrating an epilepsy surgery-induced temporal asymmetry deformity on the left side.

The degree of temporal asymmetry was calculated as the percentage difference in thickness between the right and left temporal sides (or between the operated and non-operated sides) using the formula: (longer side−shorter side)/longer side×100. A total of 192 MRI scans (96 pre-epilepsy surgery and 96 post-epilepsy surgery) were linearly ranked based on incremental asymmetry values. Temporal asymmetry was subsequently categorized into four severity levels (I, II, III, and IV), each containing 48 images, ranging from the minimum to the maximum asymmetry percentage values.

Using the temporal thickness difference data, a previously described but not validated cutoff value of 10% for temporal asymmetry (Kim et al., 2018) was applied to classify included preoperative and postoperative images into “no visible temporal asymmetry” (≤cut-off value) versus “at risk of clinically obvious temporal asymmetry deformity” (>cut-off value). This cutoff value was further utilized for comparative analysis of subjective asymmetry detection by various observer categories, including trained clinicians and untrained observers, as well as subcategories such as family members, laypeople, clinical observers, and surgical observers.

Stimuli processing

The stimulus materials used to assess the detection rate of temporal asymmetry with the subjective panel assessment instrument consisted of 3D frontal views, generated from preoperative and 12-month postoperative MRI scans. Native-space images were head-oriented using horizontal, coronal, and sagittal planes, and the aligned patient-specific 3D craniofacial soft tissue images were automatically adjusted to convert all right-sided operated regions to be consistently viewed as left-sided operated regions. The 3D models were also cropped to remove the lower craniofacial portion, particularly beneath the level of the mouth corners. Corresponding preoperative 3D craniofacial images were similarly flipped and cropped, ensuring consistency between the preoperative and postoperative images for subjective assessment. These horizontally flipped and cropped images (Fig. 2B) attenuated the unconscious bias on the part of the human observer, as the adjustments ensured that any potential visible asymmetry could be assessed without the influence of side-specific differences and confounding external facial features, such as the lower lip, oral commissures, lower cheek, mandible, and chin regions, in the images (Deall et al., 2016; Schwirtz et al., 2018).

All preoperative and postoperative 3D craniofacial images were presented at full brightness on a 15-inch MacBook Pro (Apple, Inc., Cupertino, CA, USA) using a timed Microsoft PowerPoint presentation (Microsoft Corporation, Redmond, WA, USA). The stimuli were randomly organized with varying levels of asymmetry and surgical statuses (pre- and postoperative) to minimize sequence effects. Each image was displayed for 6 s to allow observers to evaluate the asymmetry and respond to the questionnaire (Padwa, Kaiser & Kaban, 1997; Rhodes et al., 2005; Chatrath et al., 2007; Carvalho et al., 2012; Jackson et al., 2013; Lewis, 2017; Chou et al., 2019; Wan, Tsai & Lo, 2021). A blank slide was inserted between images for 2 s to reduce direct comparisons between consecutive stimuli. To mitigate fatigue during the rating task, automated breaks were incorporated into the presentation: a 1-minute break after every 13 slides and a 5-minute break after every 39 slides (Wood, Fisher & Andres, 1997; Larese Filon et al., 2019; Taylor-Phillips & Stinton, 2019; Hsiao et al., 2023).

Subjective assessment

The temporal region, identified as the area of interest, was evaluated through subjective intuitive perception by a panel of clinicians and lay observers, employing a widely used binary rating system: binomial symmetry versus asymmetry grading system (Padwa, Kaiser & Kaban, 1997; Rhodes et al., 2005; Chatrath et al., 2007; Carvalho et al., 2012; Jackson et al., 2013; Lewis, 2017; Chou et al., 2019; Wan, Tsai & Lo, 2021).

All clinicians and lay observers were instructed to assess each slide spontaneously regarding the of the temporal region, following standardized instructions. (1) The temporal anatomical subunit within the craniofacial region was defined and explained. Each observer was asked to identify this region on their own head, and no comprehension issues were observed. (2) Symmetry was described as both sides of the temporal region appearing similar, while asymmetry indicated a difference between the sides. Observers were asked to paraphrase this concept to confirm their understanding, with no issues identified. (3) A printed folder was provided to guide the use of the rating system, and observers were explicitly instructed: “Please rate the individual on the slide using the scale”.

The clinician panel comprised surgical and clinical specialists. Surgical and clinical observers included 16 external craniofacial plastic surgeons (six females) and 16 external dermatologists (13 females), respectively, each with 5 to 22 years of post-certification experience. The lay observer panel included 16 parents (eight females, aged 20–57 years) of individuals who had undergone surgery for cleft-craniofacial deformities, and 16 individuals (eight females, aged 18–52 years) without personal or familial histories of craniofacial surgery, cleft-craniofacial deformities, or plastic surgery-related experiences. These lay observers represented diverse socioeconomic and educational backgrounds (upper secondary, postsecondary, and tertiary education, classified according to the Brazilian stratification system) (Weller & Horta Neto, 2021). None had formal training in nursing, medicine, dentistry, or psychology.

All observers had normal or corrected-to-normal vision, no history of psychotropic drug use or mental impairments, no personal relationships with the patients, and were blinded to the study’s purpose except for the region of interest. They could not revisit slides, as initial impressions were prioritized, and received no financial compensation.

Reproducibility

Reproducibility was evaluated in terms of reliability and precision. All objective linear distance measurements were performed twice by the same examiner, with a 4-week interval, and their mean was used for analysis. Inter-examiner reliability was assessed by having a second examiner independently repeat measurements on 30% of randomly selected images. Intra- and inter-examiner reliability were quantified using intraclass correlation coefficients (ICC) (McGraw & Wong, 1996; Portney & Watkins, 2000; Koo & Li, 2016). Precision was evaluated through the relative error magnitude (Utermohle, Zegura & Heathcote, 1983; Ward & Jamison, 1991; Gordon & Bradtmiller, 1992).

To assess intra-observer reliability, 20% of the 3D images were randomly duplicated in the panel sets. Cohen’s Kappa (k) and Fleiss’s multirater kappa (kFleiss) were applied to evaluate intra-observer and inter-observer reliabilities, respectively, for nominal data (Fleiss, 1971; Kraemer, 2014; Asmundson, 2022).

Statistical analysis

The frequency of clinical detection of temporal asymmetry by each observer category and subcategory was calculated across varying levels of severity (I to IV), predefined cutoff values of asymmetry (0–10% and >10% asymmetry), and preoperative and postoperative image types. The clinical detection rate for each category and subcategory of observers was determined by dividing the number of asymmetry responses identified by observers in the respective category and subcategory by the total number of possible responses for that category and subcategory. A detection rate of 90% or higher was pre-established as the threshold for determining panel-based clinical recognition of obvious temporal asymmetry using the binary grading system (Padwa, Kaiser & Kaban, 1997; Rhodes et al., 2005; Chatrath et al., 2007; Carvalho et al., 2012; Jackson et al., 2013; Lewis, 2017; Chou et al., 2019; Wan, Tsai & Lo, 2021).

The data distribution was assessed using the Kolmogorov–Smirnov test. Statistical comparisons were conducted using the Kruskal–Wallis test, Z-test, Chi-square test, Fisher’s Exact Test, Wilcoxon signed-rank test, and Mann–Whitney U test as appropriate for the data types and sample structures. When applicable, relationships between variables (e.g., sex and temporal thickness difference) were evaluated using Spearman’s rank correlation coefficient (r). A Bonferroni correction was applied to adjust for multiple comparisons. A P-value of <0.05 was considered statistically significant, and all tests (IBM SPSS software, version 23.0; Armonk, NY, USA) were two-sided.

Results

Objective analysis

For objective measurements, intra- and inter-examiner reliability ranged from moderate to excellent. Regarding the error magnitude statistic, temporal thickness measurements demonstrated good precision for the REM parameter (Table S1).

Most MRI craniofacial images (61.5%; P < 0.001) exhibited temporal thickness differences of less than 10% (Table 1; Fig. 3). Across asymmetry severity levels, level I showed the lowest temporal thickness values, followed by levels II through IV (all P < 0.001). Asymmetry levels I and II had no temporal thickness differences exceeding 10%, while level III included both values above and below 10% (54.2% and 45.8% of images, respectively; P > 0.05). Level IV consisted exclusively of images with values above 10% (Table 1; Fig. 4).

Table 1 Temporal thickness difference distribution across asymmetry levels.

Temporal asymmetry level	Temporal thickness difference	P-value	
	Overall	≤10%	>10%		
Level I–IV					
n (%)	192 (100)	118 (61.46)	74 (38.54)	<0.001*	
Temporal thickness difference (%)
m ± sd (range)	11.46 ± 9.77
[1.12–46.56]	4.87 ± 2.08
[1.12–9.90]	21.96 ± 7.80
[10.19–46.56]	<0.001*	
Level I					
n (%)	48 (25)	48 (100)	0 (0)	–	
Temporal thickness difference (%)
m ± sd (range)	2.93 ± 0.99
[1.12–4.12]	2.93 ± 0.99
[1.12–4.12]	0 ± 0
[0–0]	–	
Level II					
n (%)	48 (25)	48 (100)	0 (0)	–	
Temporal thickness difference (%)
m ± sd (range)	5.35 ± 0.61
[4.13–6.52]	5.35 ± 0.61
[4.13–6.52]	0 ± 0
[0–0]	–	
Level III					
n (%)	48 (25)	22 (45.83)	26 (54.17)	0.541	
Temporal thickness difference (%)
m ± sd (range)	11.52 ± 3.54
[6.67–17.19]	8.08 ± 1.08
[6.67–9.90]	14.43 ± 1.83
[10.19–17.19]	<0.001*	
Level IV					
n (%)	48 (25)	0 (0)	48 (100)	–	
Temporal thickness difference (%)
m ± sd (range)	26.04 ± 6.66
[18.39–46.56]	0 ± 0
[0–0]	26.04 ± 6.66
[18.39–46.56]	–	
P -value	<0.001*,**	–	–	–	
Notes.

–, no data for comparison purpose.

n, number of craniofacial images.

%, percentage.

m, mean.

sd, standard deviation.

Preop, preoperative.

Postop, postoperative.

* Significant after applying the Bonferroni correction.

** Level I < Level II < Level III < Level IV, with p < 0.001 for all pairwise comparisons.

Figure 3 Box plots illustrating the distribution of temporal thickness differences.

For further details, refer to Table 1.

Figure 4 Box plots illustrating the distribution of temporal thickness differences across asymmetry levels and preoperative/postoperative image types.

Asymmetry levels I and II showed no temporal thickness differences exceeding 10%, whereas level III included values both above and below this threshold. Level IV exclusively comprised three-dimensional craniofacial images with temporal thickness differences greater than 10%. Preoperative three-dimensional craniofacial images exhibited only differences below 10%, while postoperative images encompassed values both above and below this threshold. For further details, refer to Table 2.

Postoperative images exhibited a significantly (P < 0.001) greater temporal thickness difference than preoperative images (Table 2). No preoperative image had a temporal thickness difference greater than 10% (ranging from 1.12% to 9.23%), while postoperative images displayed a broader range of temporal thickness differences, from 1.99% to 46.56% (Table 2; Fig. 4). Both preoperative and postoperative images were distributed across asymmetry severity levels I to III, while only postoperative images were categorized as level IV (all P < 0.001) (Table 2). Subanalyses revealed no significant influence (all P > 0.05) of the tested parameters (age, sex, and surgical side) on temporal thickness differences, considering preoperative/postoperative image types, asymmetry cutoff, and severity levels.

Table 2 Temporal thickness difference distribution across asymmetry levels and preoperative/postoperative image types.

Temporal asymmetry level	Temporal thickness difference	
	Overall		≤10%		>10%	
	Preop	Postop	P -value	Preop	Postop	P -value	Preop	Postop	
Level I–IV									
n (%)	96 (100)	96 (100)	–	96 (100)	22 (22.92)	<0.001*	–	74 (77.08)	
Temporal thickness difference (%)
m ± sd (range)	4.60 ± 1.92
[1.12–9.23]	18.32 ± 9.64
[1.99–46.56]	<0.001*	4.60 ± 1.92
[1.12–9.23]	6.09 ± 2.33
[1.99–9.90]	0.013	–	21.96 ± 7.80
[10.19–46.56]	
Level I									
n (%)	42 (87.5)	6 (12.5)	<0.001*	42 (87.5)	6 (12.5)	<0.001*	–	–	
Temporal thickness difference (%)
m ± sd (range)	2.85 ± 1.00
[1.12–4.12]	3.51 ± 0.70
[1.99–4.03]	0.139	2.85 ± 1.00
[1.12–4.12]	3.51 ± 0.77
[1.99–4.03]	0.139	–	–	
Level II									
n (%)	40 (83.3)	8 (16.7)	<0.001*	40 (83.3)	8 (16.7)	<0.001*	–	–	
Temporal thickness difference (%)
m ± sd (range)	5.35 ± 0.61
[4.13–6.52]	5.35 ± 0.59
[4.46–6.24]	0.989	5.35 ± 0.61
[4.13–6.52]	5.35 ± 0.63
[4.46–6.24]	0.989	–	–	
Level III									
n (%)	14 (29.2)	34 (70.8)	<0.001*	14 (29.2)	8 (16.7)	0.131	–	26 (54.1)	
Temporal thickness difference (%)
m ± sd (range)	7.69 ± 0.94
[6.67–9.23]	13.09 ± 2.95
[7.22–17.19]	<0.001*	7.69 ± 0.94
[6.67–9.23]	8.76 ± 1.02
[7.22–9.90]	0.020	–	14.43 ± 1.83
[10.19–17.19]	
Level IV									
n (%)	–	48 (100)	–	–	–	–	–	48 (100)	
Temporal thickness difference (%)
m ± sd (range)	–	26.04 ± 6.59
[18.39–46.56]	–	–	–	–	–	26.04 ± 6.66
[18.39–46.56]	
P -value	–	<0.001*,**	–	–	–	–	–	–	
Notes.

–, no data for comparison purpose.

n, number of craniofacial images.

%, percentage.

m, mean.

sd, standard deviation.

Preop, preoperative.

Postop, postoperative.

* Significant after applying the Bonferroni correction.

** Level I < Level II < Level III < Level IV, with p < 0.001 for all pairwise comparisons.

Subjective assessment

The intra-observer reliability testing demonstrated a strong level of agreement (k = 0.80–0.90) for untrained observers and an almost perfect level of agreement (k > 0.90) for trained observers. The inter-observer reliability testing showed a strong level of agreement (kFleiss = 0.80–0.90) across both subcategories and categories of observers (Table S2).

For both observer category (Tables 3 to 5) and single-observer (Table 6) analyses, clinicians detected temporal asymmetry with significantly greater frequency (all P < 0.001) than lay observers, considering overall temporal thickness difference, the 10% cutoff, preoperative/postoperative image types, and severity levels (except for level IV). In both the observer subcategory (Tables 3 to 5) and single-observer (Table 6) analyses, laypeople had the lowest detection rate (all P < 0.001), followed by family members, while surgical and clinical observers demonstrated higher detection rates for overall asymmetry, the 10% cutoff, preoperative/postoperative image types, and severity levels (except for level IV).

Table 3 Detection rates of temporal asymmetry across observer categories/subcategories and asymmetry cutoffs.

Parameters	Detection rate of temporal asymmetry (%)	
	Temporal thickness difference	P -value	
	Overall
(i = 192)	≤10%
(i = 118)	>10%
(i = 74)		
All observers (n = 64)	38.40%	3.32%	94.32%	<0.001*	
Total number of possible responses (t)	12,288	7,552	4,736	–	
Untrained observers (n = 32)	37.00%	2.41%	92.15%	<0.001*	
Total number of possible responses (t)	6,144	3,776	2,368	–	
Family members (n = 16)	37.57%	2.49%	93.50%	<0.001*	
Lay people (n = 16)	36.43%	2.33%	90.79%	<0.001*	
P-value (intra-subcategory)	0.263	0.797	<0.001*	–	
Total number of possible responses (t)	3,072	1,888	1,184	–	
Trained clinicians (n = 32)	39.79%	4.24%	96.49%	<0.001*	
Total number of possible responses (t)	6,144	3,776	2,368	–	
Clinical observers (n = 16)	39.26%	3.60%	96.11%	<0.001*	
Surgical observers (n = 16)	40.33%	4.87%	96.88%	<0.001*	
P-value (intra-subcategory)	0.3897	0.0525	0.3146	–	
Total number of possible responses (t)	3,072	1,888	1,184	–	
P -value	–	–	–	–	
Untrained versus trained observers	<0.001*	<0.001*	<0.001*	–	
Inter-subcategories	0.154	<0.001*,***	<0.001*,**	–	
Notes.

–, no data for comparison purpose.

n, number of observers.

i, number of craniofacial images.

t, total number of possible responses.

%, percentage.

* Significant after applying the Bonferroni correction.

** Lay people < family members < clinical observers = surgical observers, with p < 0.001 for all pairwise comparisons, except between clinical observers and surgical observers (p > 0.05).

*** Lay people = family members < clinical observers = surgical observers, with p < 0.001 for all pairwise comparisons, except between lay people and family members, and clinical observers and surgical observers (p > 0.05).

Table 4 Detection rates of temporal asymmetry across observer categories/subcategories, asymmetry cutoffs, and preoperative/postoperative image types.

Parameters	Detection rate of temporal asymmetry (%)	
	Overall temporal thickness difference (i = 192)	Temporal thickness difference≤10% (i = 118)	Temporal thickness difference > 10% (i = 74)	P -value #	
	Preop
(i = 96)	Postop
(i = 96)	P -value	Preop
(i = 96)	Postop
(i = 22)	P -value	Preop
(i = 0)	Postop
(i = 74)		
All observers (n = 64)	3.24%	73.55%	<0.001*	3.24%	3.69%	0.458	–	94.32%	<0.001*	
Total number of possible responses (t)	6,144	6,144	–	6,144	1,408	–	–	4,736	–	
Untrained observers (n = 32)	2.41%	71.58%	<0.001*	2.41%	2.41%	1.00	–	92.15%	<0.001*	
Total number of possible responses (t)	3072	3072	–	3072	704	–	–	2368		
Family members (n = 16)	2.54%	72.59%	<0.001*	2.54%	2.27%	0.772	–	93.50%	<0.001*	
Lay people (n = 16)	2.28%	70.57%	<0.001*	2.28%	2.56%	0.758	–	90.79%	<0.001*	
P-value (intra-subcategory)	0.677	0.197	–	0.677	0.805	–	–	0.0021	–	
Total number of possible responses (t)	1,536	1,536	–	1,536	352	–	–	1,184	–	
Trained clinicians (n = 32)	4.07%	75.52%	<0.001*	4.07%	4.97%	0.28	–	96.49%	<0.001*	
Total number of possible responses (t)	3,072	3,072	–	3,072	704	–	–	2,368	–	
Clinical observers (n = 16)	3.71%	74.80%	<0.001*	3.71%	3.13%	0.595	–	96.11%	<0.001*	
Surgical observers (n = 16)	4.43%	76.24%	<0.001*	4.43%	6.82%	0.060	–	96.88%	<0.001*	
P-value (intra-subcategory)	0.315	0.356	–	0.315	0.024	–	–	0.315	–	
Total number of possible responses (t)	1,536	1,536	–	1,536	352	–	–	1,184	–	
P -value	–	–	–	–	–	–	–	–	–	
Untrained versus trained observers	<0.001*	<0.001*	–	<0.001*	<0.001*	–	–	<0.001*	–	
Inter-subcategories of observers	<0.001*,**	<0.001*,**	–	<0.001*,**	<0.001*,***	–	–	<0.001*,****	–	
Notes.

–, no data for comparison purpose.

%, percentage.

n, number of observers.

i, number of craniofacial images.

t, total number of possible responses.

Preop, preoperative.

Postop, postoperative.

# Postoperative (≤10% of asymmetry) versus postoperative (>10% of asymmetry).

* Significant after applying the Bonferroni correction.

** Surgical observers scored significantly higher than lay people (p < 0.001), with no significant differences observed in the remaining pairwise comparisons (p > 0.05).

*** Lay people = family members = clinical observers <surgical observers, with p < 0.001 for all pairwise comparisons, except between lay people and family members, lay people and clinical observers, and family members and clinical observers (p > 0.05).

**** Lay people <family members <clinical observers = surgical observers, with p < 0.001 for all pairwise comparisons, except between clinical observers and surgical observers (p > 0.05).

Table 5 Detection rates of temporal asymmetry across observer categories/subcategories, severity levels, and asymmetry cutoffs.

Parameters	Detection rate of temporal asymmetry (%)	
	Overall temporal thickness
difference	Temporal thicknessdifference≤10%	Temporal thickness difference > 10%	
	Level I	Level II	Level III	Level IV	P -value	Level I	Level II	Level III	Level III	Level IV	
All observers (n = 64)	0.72%	4.07%	48.80%	100.00%	<0.001*,**	0.72%	4.07%	7.39%	83.83%	100.00%	
Untrained observers (n = 32)	0.59%	2.73%	44.66%	100.00%	<0.001*,**	0.59%	2.73%	5.68%	77.64%	100.00%	
Family members (n = 16)	0.65%	2.60%	47.01%	100.00%	<0.001*,**	0.65%	2.60%	6.25%	81.49%	100.00%	
Lay people (n = 16)	0.52%	2.86%	42.32%	100.00%	<0.001 *,**	0.52%	2.86%	5.11%	73.80%	100.00%	
P-value (intra-subcategory)	<0.001*	<0.001*	<0.001*	–	–	<0.001*	<0.001*	<0.001*	<0.001*	–	
Trained clinicians (n = 32)	0.85%	5.40%	52.93%	100.00%	<0.001*,**	0.85%	5.40%	9.09%	90.02%	100.00%	
Clinical observers (n = 16)	0.78%	3.26%	52.99%	100.00%	<0.001*,**	0.78%	3.26%	10.51%	88.94%	100.00%	
Surgical observers (n = 16)	0.91%	7.55%	52.86%	100.00%	<0.001*,**	0.91%	7.55%	7.67%	91.11%	100.00%	
P-value (intra-subcategory)	<0.001*	<0.001*	<0.001*	–	–	<0.001*	<0.001*	<0.001*	<0.001*	–	
P -value	–	–	–	–	–	–	–	–	–	–	
Untrained versus trained observers	<0.001*	<0.001*	<0.001*	–	–	<0.001*	<0.001*	<0.001*	<0.001*	–	
Inter-subcategories of observers	<0.001*,***	<0.001*,***	<0.001*,****	–	–	<0.001*,***	<0.001*,***	<0.001*,****	<0.001*,***	–	
Notes.

–, no data for comparison purpose.

%, percentage.

n, number of observers.

* Significant after applying the Bonferroni correction.

** Level I < Level II < Level III < Level IV, with p < 0.001 for all pairwise comparisons.

*** Lay people < family members < clinical observers < surgical observers, with p < 0.001 for all pairwise comparisons.

**** Lay people < family members < surgical observers < clinical observers, with p < 0.001 for all pairwise comparisons.

Table 6 Detection rates of temporal asymmetry by individual observers.

Single-observer analysis	Detection rate of temporal asymmetry (%)	
	Temporal thickness difference	P -value	
	Overall (i = 192)	≤10% (i = 118)	>10% (i = 74)		
All observers (n = 64) m ± sd (range)	38.40 ± 1.94 [35.42–43.23]	3.32 ± 2.03 [0.85–9.32]	94.32 ± 2.69 [90.54–98.65]	<0.001*	
Untrained observers (n = 32)	37.00 ± 0.87 [35.42–39.06]	2.41 ± 0.81 [0.85–4.24]	92.15 ± 1.66 [90.54–95.95]	<0.001*	
Family members (n = 16) m ± sd (range)	37.57 ± 0.68 [36.46–39.06]	2.49 ± 0.72 [1.69–4.24]	93.50 ± 1.23 [91.89–95.95]	<0.001*	
Lay people (n = 16) m ± sd (range)	36.43 ± 0.64 [35.42–38.02]	2.33 ± 0.90 [0.85–4.24]	90.79 ± 0.54 [90.54–91.89]	<0.001*	
P-value (intra-subcategory)	<0.001*	0.657	<0.001*	–	
Trained clinicians (n = 32)	39.79 ± 1.69 [37.50–43.23]	4.24 ± 2.45 [1.69–9.32]	96.49 ± 1.49 [93.24–98.65]	<0.001*	
Clinical observers (n = 16) m ± sd (range)	39.26 ± 1.45 [37.50–41.67]	3.60 ± 2.11 [1.69–6.78]	96.11 ± 1.47 [93.24–97.30]	<0.001*	
Surgical observers (n = 16) m ± sd (range)	40.33 ± 1.78 [37.50–43.23]	4.87 ± 2.67 [1.69–9.32]	96.88 ± 1.46 [94.59–98.65]	<0.001*	
P-value (intra-subcategory)	0.081	0.167	0.171	–	
P -value	–	–	–	–	
Untrained versus trained observers	<0.001*	<0.001*	<0.001*	–	
Inter-subcategories of observers	<0.001*,**	<0.001*,***	<0.001*,***	–	
Notes.

–, no data for comparison purpose.

n, number of observers.

i, number of craniofacial images.

%, percentage

m, mean.

sd, standard deviation.

* Significant after applying the Bonferroni correction.

** Lay people < family members < clinical observers = surgical observers, with p < 0.001 for all pairwise comparisons, except between clinical observers and surgical observers (p > 0.05).

*** Surgical observers scored significantly higher than lay people (p < 0.001), with no significant differences observed in the remaining pairwise comparisons (p > 0.05).

For images with temporal thickness difference of 10% or less, no significant (all P > 0.05) difference was observed between preoperative and postoperative image types across observer categories and subcategories (Table 4). As asymmetry severity level increases, the detection rate of asymmetry by all observer categories and subcategories also increases significantly (all P < 0.001), considering both overall temporal thickness difference and the 10% cutoff (Table 5). For level III severity, clinical observers had the highest detection rates in the overall temporal thickness difference parameter and when appraising images with temporal thickness differences of 10% or less (all P < 0.001) (Table 5). Surgical observers had the highest detection rates for levels I and II (regardless of temporal thickness difference) and for level III when appraising images with temporal thickness differences greater than 10% (all P < 0.001) (Table 5). For level IV asymmetry, all observer categories and subcategories detected asymmetry in all 48 images, regardless of the overall temporal thickness difference or the 10% cutoff (all P > 0.05) (Table 5). Subanalyses revealed no significant influence (all P>0.05) of the tested parameters (age, sex, and surgical side) on detection rates, considering temporal thickness difference, observer categories/subcategories, preoperative/postoperative image types, asymmetry cutoff, and severity levels.

Threshold-based detection of asymmetry

The detection rate of temporal asymmetry in 3D craniofacial images with temporal thickness difference greater than 10% (range: 10.186% to 46.555%) was significantly higher than in images with temporal thickness difference of 10% or less (range: 1.118% to 9.901%), across both observer categories and subcategories (all P < 0.001) (Table 3; Fig. S1). Temporal thickness differences greater than 10% were clinically detected with a frequency exceeding 90% by both observer categories (trained clinicians and untrained lay observers) and subcategories (family members, lay people, and clinical and surgical observers) (Tables 3, 4 and 6; Fig. S2). The clinical detection rate for 3D craniofacial images with temporal thickness differences less than 10% (range: 1.118% to 9.901%) did not exceed 90% for any observer category or subcategory (Table 3). No significant correlation (all P > 0.05) was found between age, sex, surgical side, and the asymmetry cutoff parameter, considering temporal thickness difference, observer categories/subategories, preoperative/postoperative image types, or severity levels.

Discussion

Currently, there is no gold standard for identifying temporal region asymmetry (Wang et al., 2017; Laloze et al., 2019; Shay et al., 2022; Nasim et al., 2024). To address this, instead of computer-generated asymmetrical models (e.g., chimeras or blends) (Naini et al., 2012a), we used unaltered 3D craniofacial models from surgically treated patients, preserving the natural clinical context and reflecting real-world epilepsy care in a Brazilian population of mixed ethnic backgrounds (Giacomini et al., 2020; Secolin et al., 2021). By applying appropriate statistical tests and established cut-off values (Padwa, Kaiser & Kaban, 1997; Rhodes et al., 2005; Chatrath et al., 2007; Kottner et al., 2011; Carvalho et al., 2012; Jackson et al., 2013; Lewis, 2017; Chou et al., 2019; Wan, Tsai & Lo, 2021), both the subjective panel assessments and the objective measurements of temporal soft tissue thickness differences met criteria for reliability and precision, supporting the robustness of the data collected for the temporal region.

Our analyses showed that 3D craniofacial images with prior temporal surgery exhibited greater asymmetry than those without surgical intervention, and perfect symmetry was absent in all images. These results support the presence of craniofacial asymmetry across the population (Chou et al., 2019; Crins-de Koning et al., 2025) and indicate that surgery-induced asymmetry is more clinically noticeable (Cheong & Lo, 2011). Our data reflect natural variability in the temporal region under real-world conditions, providing a useful reference for future studies on both natural and surgical asymmetry. To broaden these findings, future MRI-based research could include diverse ethnic groups and patients undergoing other temporal surgeries (e.g., for trauma, aneurysms, tumors) and assess additional asymmetry indices. Studies might also examine the influence of underlying brain and cranial structures on craniofacial asymmetry (Marečková et al., 2013; Kong et al., 2018).

In our study, subjective judgments of temporal asymmetry followed a consistent pattern: larger temporal thickness differences associated with higher detection rates. Both trained and untrained observers performed similarly for severe asymmetry (level IV), but detection rates varied significantly for milder asymmetries (levels I–III), with laypeople showing the lowest rates, followed by family members, while clinical and surgical observers demonstrated higher accuracy. Overall, lay observers detected asymmetry less frequently than clinicians.

Consistent with previous studies (Kokich, Kokich & Kiyak, 2006; An et al., 2014), specialized clinicians demonstrated superior recognition of asymmetry, likely due to their ability to apply clinical skills to subtle asymmetries. The clinicians in our study—craniofacial plastic surgeons and dermatologists experienced in treating appearance-altering craniofacial conditions—possess extensive expertise in managing temporal region asymmetries, including the use of fillers and grafts for reconstructive and aesthetic purposes. Unlike untrained observers, clinicians develop enhanced visualization, judgment, and sensitivity through rigorous training and clinical practice, which may explain their higher detection rates. We also hypothesize that clinicians’ superior performance may reflect conscious or unconscious motivational bias (Montibeller & von Winterfeldt, 2015), as they may invest additional effort in identifying asymmetry in regions closely related to their professional expertise.

Interestingly, studies show that laypeople with exposure to a specific condition—for example, individuals who have undergone orthodontic treatment—are better at perceiving asymmetry, such as incisal plane canting or midline shifts, than those without such experience (Naini et al., 2012a; Naini et al., 2012b; An et al., 2014). Similarly, managing cleft-craniofacial conditions poses significant challenges for patients and families, and family members involved in long-term care may develop greater awareness and critical perception of craniofacial symmetry (Denadai & Lo, 2022). This may explain why, in our study, family members demonstrated higher recognition rates of asymmetry compared to laypeople without prior exposure to appearance-altering craniofacial conditions. Our findings suggest that future research should consider subdividing lay observers rather than treating them as a single group. To further explore how technical and experiential backgrounds influence temporal asymmetry assessment, future studies could examine whether other trained healthcare providers (e.g., neurologists, neurosurgeons, psychologists managing epilepsy) and different untrained observers (e.g., family members of patients with or without epilepsy surgery) perceive asymmetry differently. Additionally, research should investigate how various skills and strategies—such as perceptual fluency, cultural influences, artistic experience, focus on specific facial regions, and sensitivity to shape, contour, or light and shadow (Pinheiro et al., 2023)—affect the perception of symmetry versus asymmetry in 3D craniofacial images across different observer groups. However, it should be acknowledged that visually assessing temporal asymmetry, particularly for untrained observers, remains inherently challenging and represents a limitation of such studies.

Similar to many medical and nonmedical contexts (Bathiany, Hidding & Scheffer, 2020; Nakajima, Okuda & Komatsu, 2021), the perception of craniofacial asymmetry appears to follow a threshold model, becoming clinically noticeable only beyond a certain degree of deviation (Parrini et al., 2016; Wang et al., 2017). Studies have examined various craniofacial subunits to define objective cutoff values—using linear, angular, or percentage measures—that correspond to subjective detection by clinicians and laypeople ((Padwa, Kaiser & Kaban, 1997; Parrini et al., 2016; Wang et al., 2017; Lee, Dumrongwongsiri & Lo, 2019)). Each craniofacial subunit has a unique perceptual threshold that distinguishes normal variation from clinically recognizable deviations (Padwa, Kaiser & Kaban, 1997; Parrini et al., 2016; Wang et al., 2017).

To date, no study has quantified the minimal degree of objectively measured temporal asymmetry detectable by both trained and untrained observers. Although a 10% threshold has been proposed for defining visible unilateral temporal deformity (Kim et al., 2018), this value remains unvalidated through formal observer detection analyses. Craniofacial soft tissue thickness—a linear measurement widely used in clinical and research settings for purposes such as evaluating surgery-induced changes, assessing sarcopenia, serving as a prognostic marker in cancer, and aiding facial approximation in forensic medicine (Stephan & Devine, 2009; Hona & Stephan, 2024)—was therefore employed as the objective measure in the current study.

Our cumulative data demonstrated that both clinicians and lay observers were able to identify temporal asymmetry at a threshold of 10% difference in temporal thickness. All observer categories and subcategories met or exceeded the minimum clinical recognition level, with over 90% accuracy in detecting asymmetry in 3D craniofacial images exhibiting temporal thickness differences greater than 10%. This threshold effectively distinguished between 3D craniofacial images predicted to be unaffected (preoperative images) and those hypothesized to exhibit visible temporal asymmetry (postoperative images). Based on our findings, and in alignment with definitions supported by previous research (Kim et al., 2018), a 10% difference in temporal thickness can be proposed as a discriminative threshold for the clinical detection of temporal asymmetry by both trained and untrained observers. Importantly, differences in sensitivity for the detection of temporal asymmetry between clinicians and lay observers could have practical implications. In real-world settings, lower sensitivity among lay observers might mean that minor temporal asymmetries, although noticeable to clinicians, remain imperceptible to the public, potentially reducing patients’ psychosocial distress or perceived need for intervention. Therefore, recognizing this potential clinical scenario could help avoid unnecessary treatment or overinflated diagnoses driven by clinicians’ higher sensitivity. Understanding these dynamics is crucial for guiding patient counseling, setting realistic expectations, and informing decisions about therapeutic interventions.

While our study does not answer all questions about craniofacial asymmetry, it offers valuable insights into how clinicians and lay observers recognize temporal asymmetry. We have also established a key threshold for clinical detection, which may benefit interdisciplinary epilepsy care. This data provides an initial framework for counseling adult patients and setting expectations for potential postsurgical asymmetry. Moreover, it could aid perioperative assessments in temporal reconstruction (Vaca et al., 2017; Gonçalves et al., 2021), where residual asymmetry may exist but remain imperceptible in interpersonal interactions.

Potential caveats of this study should be acknowledged. This was not an epidemiological investigation, and we did not assess the prevalence of temporal deformity, explore multiple levels of nonindependence, or investigate potential predictors of its occurrence following epilepsy surgery. The study was also not designed to examine hierarchical data structures or to model clustered or nested sources of variability. These aspects merit future research using alternative methodologies, including mixed-effects model analyses. Future investigation could explore specific structural components of the temporal region, such as fat, muscle, and bone, to provide a more detailed understanding of asymmetry. Studies could also assess the severity of asymmetry and treatment needs using continuous or ordinal rating scales, as well as additional parameters such as surface area and volume, allowing for more nuanced differentiation beyond binary classification. Additionally, leveraging artificial intelligence or machine learning techniques for automated detection holds promise for matching or even exceeding human performance in identifying temporal asymmetry.

Conclusions

This study identified a 10% difference in temporal thickness as the threshold for the clinical recognition of temporal asymmetry by both trained clinicians (surgeons and clinical specialists) and untrained observers (family members and laypeople). Moreover, significant differences were observed between these groups, with trained clinicians detecting asymmetry more frequently than untrained observers across most conditions—including overall temporal thickness differences, the 10% cutoff, preoperative/postoperative image types, and varying severity levels—except at the highest severity level, where all observers achieved complete detection.

Supplemental Information

Supplemental Information 1 STROBE Checklist

Supplemental Information 2 Raw data

All independent and dependent variables were adopted for the reported analysis.

Supplemental Information 3 Box plots

Box plots illustrating the distribution of detection rates of temporal asymmetry by individual observers based on three-dimensional craniofacial images with temporal thickness differences below 10%. For further details, refer to Table 6.

Supplemental Information 4 Box plots

Box plots illustrating the distribution of detection rates of temporal asymmetry by individual observers based on three-dimensional craniofacial images with temporal thickness differences greater than 10%. For further details, refer to Table 6.

Supplemental Information 5 Reproducibility statistics for temporal thickness measurement

Supplemental Information 6 Reliability statistics for panel assessment

Reliability statistics for panel assessment .

We would like to thank the panel of observers and examiners who provided help during data collection.

Additional Information and Declarations

Competing Interests

Author Contributions

Ethics

Data Availability

The authors declare there are no competing interests.

Rafael Denadai conceived and designed the experiments, performed the experiments, analyzed the data, prepared figures and/or tables, authored or reviewed drafts of the article, and approved the final draft.

Marina Koutsodontis Machado Alvim conceived and designed the experiments, performed the experiments, analyzed the data, authored or reviewed drafts of the article, and approved the final draft.

Yeonah Kang conceived and designed the experiments, performed the experiments, analyzed the data, prepared figures and/or tables, authored or reviewed drafts of the article, and approved the final draft.

Junior Chun-Yu Tu conceived and designed the experiments, performed the experiments, analyzed the data, prepared figures and/or tables, authored or reviewed drafts of the article, and approved the final draft.

Brunno M. de Campos conceived and designed the experiments, authored or reviewed drafts of the article, and approved the final draft.

Enrico Ghizoni conceived and designed the experiments, authored or reviewed drafts of the article, and approved the final draft.

Helder Tedeschi conceived and designed the experiments, authored or reviewed drafts of the article, and approved the final draft.

Clarissa Yasuda conceived and designed the experiments, prepared figures and/or tables, authored or reviewed drafts of the article, and approved the final draft.

Fernando Cendes conceived and designed the experiments, authored or reviewed drafts of the article, and approved the final draft.

The following information was supplied relating to ethical approvals (i.e., approving body and any reference numbers):

The University of Campinas (UNICAMP) granted Ethical approval to carry out the study (CAAE: 93412318.0.0000.5404).

The following information was supplied regarding data availability:

The raw measurements are available in the Supplementary File.

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
