# Peer review of "Detecting temporal asymmetry after epilepsy surgery: a 3D MRI-based comparative outcome study of clinicians and lay observers"

_PeerJ, doi:10.7717/peerj.20201_

## Round 0.1 · original submission · Major Revisions

Both reviewers considered the study meaningful and offered detailed suggestions for improvement. Please revise the manuscript accordingly, with particular attention to enhancing the clarity and precision of the methodology, and streamlining the introduction and discussion sections. Once all reviewer comments have been thoroughly addressed, please resubmit your study for further consideration.

Reviewer 1 ·

Basic reporting

1. Terminology and Audience Consideration: Given the likely clinical or technical expertise of the readership, some explanations of terminology (e.g., under "reproducibility" or in the introduction when defining "asymmetry") could be made more concise.

2. Introduction and Discussion Structure: These sections can be shortened and made more concise. It currently includes verbose and complex sentences, and some points are repeated across multiple paragraphs.

Experimental design

1. MRI Acquisition Parameters: Were the pre- and postoperative MRI scans acquired using the same scanner and with comparable image resolution or slice thickness?

2. MRI Preprocessing Steps: The manuscript would benefit from a more detailed explanation of the MRI preprocessing pipeline. Specifically, steps such as field bias correction, intensity normalization, and image registration should be described. Additionally, it is unclear how temporal thickness was measured—was a surface-based approach used?

3. Statistical Modeling: Have you accounted for the subject as a random effect when comparing pre- and postoperative temporal thickness? If so, please specify whether a mixed-effects model was used to control for repeated measures.

4. Asymmetry Threshold Justification: They referred to the use of a 10% asymmetry threshold (lines 237–240), described as "previously reported but not validated." A brief justification for its use and a discussion of how it may influence interpretation should be provided.

Validity of the findings

1. Observer Training Consistency: While observers are categorized as trained or untrained based on their professional background, it is unclear whether any calibration or practice sessions were conducted before data collection. Were any steps taken to ensure consistency in subjective scoring?

·

Basic reporting

- The article is well-written in clear, unambiguous English.
- The introduction and study objectives are adequately described, though they could be made more concise for improved readability (see Additional Comments for details).
- The manuscript is well-supported by relevant literature. However, only 40 of the 142 cited references are from the past five years. The authors are encouraged to update the reference list with more recent and relevant studies or to include a statement addressing the limited availability of recent literature in this field.
- The article structure adheres to PeerJ's formatting and submission requirements. The abstract effectively summarizes the research and its key findings.
- Tables and figures are relevant, well-designed, and of high quality.
- The supplementary data provided are appropriate and sufficient to support the study’s findings.

Experimental design

- The study falls within the scope of PeerJ (Medical/Health Sciences).
- The research design is generally well-articulated in the methods section; however, certain areas would benefit from further clarification and elaboration (see Additional Comments).

Validity of the findings

- The findings are presented clearly and accurately.
- In the conclusion, beyond highlighting the significance of the 10% asymmetry threshold, the authors should underscore the consistent superiority of trained observers over untrained ones across all levels of asymmetry. This is an important distinction with implications for the method’s application in both clinical and non-clinical settings.

Additional comments

This manuscript presents a thorough and well-articulated study that effectively connects its findings to existing literature and discusses relevant clinical applications. However, the manuscript is quite lengthy (approximately 7,300 words) and would benefit from structural tightening and clarification across several sections to enhance clarity, conciseness, and overall readability.

Introduction
The introduction would benefit from being restructured using Lingard’s “Problem/Gap/Hook” heuristic to better engage the reader and streamline the argument. Specific suggestions include:
• Paragraphs 1–3 (Lines 52–96): These could be more concise, focusing primarily on defining symmetry and asymmetry, highlighting the importance of the temporal region, and outlining the clinical implications. [Problem]
• Paragraphs 4–7 (Lines 97–163) and Paragraph 8 (Lines 164–183): These sections should be condensed to present a clearer overview of the methods for detecting asymmetry, the lack of standardized criteria, and the clinical need for such standards. [Gap]
• Paragraph 9 (Lines 184–188): This section clearly articulates the study’s aim and appropriately addresses the gap identified. [Hook]

Method
The use of a cross-sectional study design is appropriate for the stated objectives. However, this section would benefit from additional methodological detail to improve transparency and reproducibility:
• Sample Size and Sampling Method: The section should include a clearer explanation of how the sample size was determined, the sampling method used, and the sampling period. These additions would greatly enhance methodological rigor.
• Operational Definition (Lines 230–243): While the classification of temporal asymmetry into percentile-based categories provides some insight, categorizing asymmetry levels based on absolute percentage differences (e.g., >10–20%, >20–30%, etc.) may provide a more intuitive framework. This could also improve clarity around diagnostic thresholds and observer sensitivity.
• Stimuli Processing (Lines 260–268): The chosen exposure protocol (6 seconds/image, 2-second blank, 1-minute break every 13 slides, 5-minute break after 39 slides) is adequately described but would benefit from a rationale. Was it based on prior studies, pilot testing, or practical constraints? Justifying these choices would enhance reader confidence in the protocol's suitability.
• Rating System (Lines 282–284): The observer rating system for assessing temporal asymmetry should be described in greater detail, including its scale, criteria, and standardization procedures.
• Observer Limitations: The inherent difficulty of visually assessing temporal asymmetry—especially for untrained observers—should be acknowledged as a limitation. Future studies could explore whether the identified thresholds are reliably recognized by both experts and laypersons

Results
• Detection Thresholds (Lines 402–414): The authors report that asymmetry detection rates increased significantly when differences exceeded 10%, aligning with Kim et al. (2018). This finding would be strengthened by including precise cutoff values for each subgroup and comparing them to previously reported thresholds. Doing so would enhance the context and generalizability of the findings.

Discussion
The discussion section contains several redundancies that repeat information already covered in earlier sections. Summarizing or removing these duplications could significantly improve the manuscript’s readability:
• Lines 417–434: These points were already covered in the background and can be made more concise.
• Lines 440–445: The study’s purpose is reiterated here, but is already clearly stated in the introduction (Lines 184–186).
• Lines 449–453: The sample size determination discussion would be more appropriately placed in the methodology section. Consider expanding this point in the methods section (Lines 195–198) and removing or simplifying it in the discussion.
• Lines 446–468 and 485–511: The rationale and methodology are discussed again here; these sections could be streamlined to avoid repetition.
• Lines 602–625: This discussion of the 10% threshold duplicates content already covered in the methods and should be condensed.
In Lines 616–625, the authors state that both clinicians and lay observers can identify temporal asymmetry when the difference exceeds a 10% threshold. This section could be strengthened by further emphasizing the practical implications of this cutoff in the general population. If lay observers demonstrate lower sensitivity to temporal asymmetry compared to clinicians, what are the potential consequences in real-world settings? Addressing this question would enhance the clinical relevance and applicability of the findings.

Conclusion
This section is concisely written and well-articulated. However, the depth of the analysis could be further enhanced by addressing the additional research questions posed in the article. Specifically, it would be valuable to elaborate on whether there are significant differences in the ratings between trained clinicians and untrained observers, and whether incremental levels of asymmetry are reflected in corresponding changes in subjective ratings. Expanding on these aspects would provide a more comprehensive understanding of observer sensitivity and the potential clinical implications of the findings.

---

## Round 0.2 · accepted · Accept

All reviewers' concerns have been adequately addressed, and the manuscript can be published in its current form.

Reviewer 1 ·

Basic reporting

The authors have thoroughly and satisfactorily addressed my comments. I have no more comments.

Experimental design

none

Validity of the findings

none

Additional comments

none

·

Basic reporting

No comment

Experimental design

No comment

Validity of the findings

No comment

Additional comments

The authors have carefully and thoughtfully addressed all the requested revisions in the current version of the manuscript. I find the revisions satisfactory, and therefore, I recommend this article for acceptance.